# Age Effects in Emotional Memory and Associated Eye Movements

**DOI:** 10.3390/brainsci12121719

**Published:** 2022-12-15

**Authors:** Daphne Stam, Laura Colman, Kristof Vansteelandt, Mathieu Vandenbulcke, Jan Van den Stock

**Affiliations:** 1KU Leuven, Leuven Brain Institute, Neuropsychiatry, 3000 Leuven, Belgium; 2Geriatric Psychiatry, University Psychiatric Center KU Leuven, 3000 Leuven, Belgium

**Keywords:** emotion recognition memory, eye-scanning patterns, total fixation duration, fixation count

## Abstract

Mnemonic enhanced memory has been observed for negative events. Here, we investigate its association with spatiotemporal attention, consolidation, and age. An ingenious method to study visual attention for emotional stimuli is eye tracking. Twenty young adults and twenty-one older adults encoded stimuli depicting neutral faces, angry faces, and houses while eye movements were recorded. The encoding phase was followed by an immediate and delayed (48 h) recognition assessment. Linear mixed model analyses of recognition performance with group, emotion, and their interaction as fixed effects revealed increased performance for angry compared to neutral faces in the young adults group only. Furthermore, young adults showed enhanced memory for angry faces compared to older adults. This effect was associated with a shorter fixation duration for angry faces compared to neutral faces in the older adults group. Furthermore, the results revealed that total fixation duration was a strong predictor for face memory performance.

## 1. Introduction

Emotional cues typically enhance episodic memory and this effect has been labeled emotional enhanced memory (EEM) [1,2]. EEM has been consistently reported for negative events [3] and has mostly been reported in immediate recognition paradigms, with short intervals between encoding and recognition. Memory consolidation refers to the transformation from an unstable to a stable status of a memory item, dependent on time [4]. It has been argued that EEM is mediated by increased processing resources during encoding and is moderated by consolidation [5,6]. Furthermore, there is accumulating evidence that EEM is affected by age [7]. In particular, EEM shows a valence-dependent age trend, with the enhancing effect of emotionally negative events decreasing from adolescence into old age [3]. For instance, when young, middle-aged, and older adults are presented with naturalistic pictures depicting scenes with a negative or neutral emotional content, younger adults recall significantly more negative than neutral images. This effect is reduced in middle-aged adults and almost absent in older adults [8]. Here, we combined eye tracking with a subsequent face recognition memory task in adolescents and older adults to investigate the interaction among each of these three factors: attentional processing during encoding, consolidation, and age.

An ingenious method to study visual attention for emotional stimuli is eye tracking. Eye tracking has the advantage of providing information about attentional mechanisms that reflect individual differences relevant to emotion regulation. Eye movements play a functional role in recognition memory of face stimuli, as restricted viewing (fixated in a single central location) during encoding results in a decline in recognition performance compared to free viewing [9]. Furthermore, there is evidence that age-related differences in face recognition memory may be related to the changes in eye-scanning behavior when viewing new faces. More transitions and increased sampling of facial features were observed in older adults [10]. A recent study revealed that older adults showed decreased engagement with the eye region and a bias to look more at the mouth of faces [11]. An eye-tracking study further assessed the age-related attentional differences using emotional face stimuli to investigate the role of visual attention [12]. The results showed an attentional preference away from angry faces in older adults. In addition to emotion, there is also evidence that aging affects mnemonic processing of social cues, although the categorical specificity of this effect remains an open question [13]. To address this issue, we included a non-face condition to our protocol, i.e., houses. Houses constitute an object category that has been extensively explored in other studies and associated with dedicated brain regions [14,15,16,17], similar to faces [18]. Face and house stimuli are further distinguished by the extent of holistic processing [19], which makes it interesting to not only study possible memory enhancement of emotional stimuli but also of social stimuli. Previous research revealed age-related effects in face memory [20,21,22]. The present design makes it possible to study age-related differences for emotion recognition and face recognition compared to non-face recognition.

In line with previous studies suggesting age-related attentional differences may cause different gaze patterns, we hypothesized that eye-scanning patterns would exert an influence on memory outcome within the different age groups. In the present study, we investigated how eye-scanning behavior during encoding is related to recognition memory for emotion by presenting young adults and older adults with emotional and neutral face stimuli. Immediately after the encoding, participants performed an immediate recognition memory test, and a delayed recognition test was performed 48 h after the encoding. Eye movements were analyzed concerning the duration of fixation in particular areas of interest (areas of interest: face, eyes, nose, mouth, and house) and the number of fixations in these areas. We further assessed eye-scanning behavior during face processing that is associated with age differences in successful subsequent recognition.

Based on the previous literature, we expected enhanced memory for faces with negative expressions in the young adults group but not in the older adults group. Based on these predictions we expected to find an association between eye-scanning patterns (fixation) and memory outcomes. In addition, we analyzed correlations between memory performance and eye tracker data for both groups separately (young adults and older adults).

## 2. Materials and Methods

### 2.1. Participants

Forty-one subjects participated in our study. They were recruited by advertisements for participation in an eye-tracker memory experiment. Participants did not receive financial compensation for their participation. Inclusion criteria consisted of (1) 18–30-year age range (young adults group) or 50–90-year age range (older adults group) and (2) an MMSE score above 25.

The young adults and older adults group consisted of 20 participants [7 males (35%); mean age ± SD = 22 ± 2 years, range 18–29] and 21 participants [9 males (43%); mean age ± SD = 69 ± 7 years, range 53–87], respectively. One participant from the older adults group was not included in the eye movement analysis due to technical issues. Participants completed the Addenbrooke’s Cognitive Examination III (ACE-III), which includes the Mini–Mental State Examination (MMSE). All participants had an ACE-III score above 71. A score of 71 is used as a cut-off for differentiating dementia from controls based on the maximum, with an acceptable sensitivity (0.87) and high specificity (0.97) [23]. In addition, all participants scored above or equal to 27 on the MMSE. Demographic data can be found in Table 1.

Chi-square test showed no significant group differences for gender (*p* = 0.606). Independent sample *t*-test showed a significant group difference for ACE-III (*p* = 0.024).

### 2.2. Experimental Design and Stimuli

In order to study emotional memory, we compared angry faces versus neutral faces. Angry faces were used as the emotional stimuli, based on our pilot study. In total, 13 healthy subjects [6 males (46%); mean age ± SD = 36 ± 15 years, range 19–50] participated in the pilot study.

For the pilot study, 200 images displaying natural facial expressions were selected from our previous studies [24,25] and validated datasets [26,27]. The stimuli displayed middle-aged women and men. The facial expression of faces in the FACES databases are rated by a total of 154 young, middle-aged, and older women and men, and only stimuli with a high intensity of facial expression were preserved within the database [26]. The Radboud Faces Database was rated by 276 women and men and reported an intraclass correlations (1, k) for intensity (0.83), clarity (0.83), and valence (0.94) [27].

The stimuli were equally divided over 5 emotion conditions: anger, fear, happiness, sadness, and neutral, i.e., 40 images for every emotion. Half of the stimuli of each category displayed males. The images were edited by removing all non-face information and presented one by one in a random order for 3000 ms, followed by a blank screen presented for 3000 ms, during which the subject was instructed to categorize the emotion in a five-alternative forced-choice task. The response alternatives were continuously presented during the 3000 ms response phase. The pilot study was preceded by three practice trials to familiarize the participant with the procedure. Based on the results of the pilot study, we selected 14 stimuli for every emotion (7 male) that were on average accurately categorized by at least 12 of the 13 subjects. This added up to a total of 140 face stimuli. In the validation study, 20 healthy psychology students from the University of Tilburg participated [3 males (15%); mean age ± SD = 20 ± 2 years, range 18–24]. A block consisted of an encoding phase during which 70 of the 140 stimuli were presented one by one against a white background for 500 ms with a 1000 ms interstimulus interval. The subjects were instructed to memorize the stimuli in furtherance of a subsequent memory test. The recognition phase immediately followed the encoding phase. In the recognition phase, all 140 stimuli were presented one by one. Viewing time was unlimited and ended by the response. Participants were instructed to indicate by a button press if the stimulus was also presented in the encoding phase. There were five response alternatives: “definitely not”; “probably not”; “no idea”; “probably yes”; and “definitely yes”. There was a 1000 ms inter-trial interval in the recognition block. We performed signal detection analysis developed for confidence rating data and calculated subject and condition-specific d’ as a parameter estimate of detection sensitivity [28]. Parameter estimates for the experimental paradigm were processed via the program R-Score Plus [29]. The receiver operating characteristic (ROC) curve in Figure 1 shows that participants recognize angry faces best ($\bar{x}$ = 1.42, SD = 0.41), then fearful faces ($\bar{x}$ = 1.08, SD = 0.45), sad faces ($\bar{x}$ = 1.02, SD = 0.52), neutral faces ($\bar{x}$ = 0.98, SD = 0.62), and finally happy expressions were remembered the least ($\bar{x}$ = 0.77, SD = 0.39). Two-sample *t*-test showed that the recognition for angry faces deviated the most from the recognition of neutral faces (*p* = 0.002).

#### 2.2.1. Stimuli

The stimuli consisted of 80 pictures of houses, which were selected from our own database and stripped of visual background. In total, 40 (20 male) neutral and 40 (20 female) angry faces were selected from our own validated database and other validated face stimuli databases [26,27,30] and stripped of visual background. All stimuli were resized to 400 pixels in height.

#### 2.2.2. Procedure

The experiment consisted of an encoding phase, an immediate recognition (IR) phase, and a delayed recognition (DR) phase (Figure 2). The encoding phase consisted of two blocks, with a duration of 2.40 min each. In every session, 80 stimuli (40 houses and 40 faces) were presented for 1500 ms against a white background and separated by a 1000 ms ISI during which a black fixation cross was presented. The second encoding session was identical to the first, except for the order of stimulus presentation. Prior to the encoding phase, participants were instructed to memorize the stimuli of the encoding phase in order to recognize them in a subsequent recognition test. Stimuli were presented on a 344.23 mm × 193.54 mm widescreen monitor (1366 × 768 dpi, 15.6”) equipped with the Tobii eye-tracking system. The screen was viewed from a distance of approximately 65 cm under free viewing conditions. The stimuli were presented at a visual angle between 5.29° and 7.04° vertically and between 5.29° and 7.04° horizontally. Before data acquisition, we created areas of interest within Tobii Studio: mouth, nose, and eyes (Figure 3a). The fixation cross was presented in the center of the screen. Relative to the face stimuli, the fixation cross was positioned around the midpoint of the virtual line connecting the nasion and nasal septum (Figure 3b). A five-point-calibration was performed prior to the test to calculate the exact eye position for each participant.

The IR phase directly followed the encoding phase. In the IR phase, the stimuli consisted of the 80 stimuli from the encoding phase intermixed with an additional 40 distractors: 20 houses and 20 faces. The procedure in the IR phase consisted of presentation of a stimulus (1500 ms) followed by a response screen (3000 ms). The response screen displayed the question “Was this picture presented in the encoding phase?” with four boxes below referring to four response alternatives: “definitely not”; “probably not”; “probably yes”; “definitely yes”. In the center of the response screen, an “X” was presented. Participants were instructed to move the “X” to the left or right by pressing the corresponding button on a laptop. A fixation screen (500 ms) followed the response screen, after which the next trial started. The 120 trials (80 targets + 40 distractors) were equally divided over four sessions of 2.46 min each and each comprising 30 trials: 15 houses (10 from the encoding phase) and 15 faces (10 from the encoding phase). In none of the encoding or recognition blocks were there more than three consecutive stimuli of the same category (angry face, neutral face, or house).

The DR was conducted 2 days after the first session, and the mean lag between encoding and DR was 47 h and 47 min (SD: 1.39). The DR is identical to the IR, except for the stimulus presentation order and the distractor stimuli, i.e., 40 new distractors were presented in the DR in order to minimize source-recognition difficulties.

Both recognition phases began with five practice trials with car stimuli, which were included to familiarize the participants with the response procedure. See also Stam et al., 2021 and Figure 2 for a schematic design of the procedure [30]. During the recognition phase, pictures were presented on a laptop running PRESENTATION^®^ 19.0 (Neurobehavioral Systems, San Francisco, CA, USA) to control stimulus presentation and response registration.

### 2.3. Eye Tracker and Eye Movement Recordings

Eye movement data were collected during the encoding phase at a sampling rate of 120 Hz using the Tobii eye tracker TX300 [31] and processed with Tobii Studio 3.4.7.

During recording, the eye tracker collects raw eye movement data points, which are processed into fixations and used to calculate eye-tracking metrics, by applying a fixation filter to the data. We applied default settings, including the Tobii fixation filter, with a velocity threshold of 0.84 pixels/ms (35 pixels) and a distant threshold (distance between two consecutive fixations) of 35 pixels (default). In short, peak values are identified, i.e., the values that are greater than both of its two closest neighbors. The list of peaks is then processed into fixations, where the start and end points of a fixation are set by two consecutive peaks. The spatial positions of the fixations are calculated by taking the median of the unfiltered data points in that interval. Secondly, the Euclidean distances between all the fixations are calculated and if the distance between two consecutive fixations falls below a second user defined threshold, the two fixations are merged into a single fixation. The process is repeated until no fixation points are closer to each other than the threshold. A detailed description of the Tobii fixation can be found in the Tobii Studio user manual (https://www.tobiipro.com/siteassets/tobii-pro/user-manuals/tobii-pro-studio-user-manual.pdf, accessed on 2 January 2019).

### 2.4. Statistical Analysis

#### 2.4.1. Behavioral Analyses

Behavioral results were analyzed according to signal detection theory [32]. R-Score Plus [29] was used to calculate d’ for confidence rating designs. D’ was calculated as a function of category (face vs. house), emotion (angry vs. neutral), interval (IR vs. DR), and group (older adults and young adults). We calculated the mean interval between the encoding phase and DR (lag) for every participant.

To evaluate the anticipated outcomes for group differences in d’ in the IR phase, we performed the following general multivariate regression model, which takes repeated measures within subjects into account. Let Y_i_ be a vector with repeated measures for the *i*^th^ subject (i…N). This general multivariate regression model assumes that Y_i_ satisfies the following regression model: Yi = Xiβ + εi with X_i_ being a matrix of covariates (e.g., intercept, group, emotion condition, and group x emotion condition), β is a vector of regression coefficients, and ε_i_ is a vector of error components with εi∼N(0, Σ). For the variance/covariance structure Σ of each subject, we considered a compound symmetry and unstructured variance/covariance matrix. Selection of the adequate variance/covariance matrix was based on a likelihood-ratio test. Reference coding was used for group (2 levels: older adults = 1 vs. young adults = 0) and emotion (2 levels: neutral = 1 vs. angry = 0). To evaluate main and interaction effects, Bonferroni-corrected post hoc tests were used. It may be noted that this model is a special case of a linear mixed model and that the mean structure X_i_β (the parameters of interest) can be interpreted as that in a classical ANOVA or regression model [33].

Second, we performed a similar model but with category (2 levels: neutral = 1 vs. angry = 0) instead of emotion as predictor. These analyses were performed for the two different memory stages (IR and DR) separately.

Lastly, we performed a similar model but with intervals (2 levels: IR = 1 vs. DR = 0) for the different conditions (house, face, angry face, neutral face) separately. Finally, a similar model was used with groups (2 levels: older adults = 1 vs. young adults = 0), intervals (2 levels: IR = 1 vs. DR = 0), and group x interval as predictors. All analyses were performed in SPSS [34].

#### 2.4.2. Eye-Tracker Analyses

Eye movement data were calculated for house, face, and the three areas of interest: mouth, nose, and eyes. For every participant, two indices for eye movement data were recorded: total fixation duration and fixation count. Total fixation duration represents the total time of fixation as it measures the sum of the duration (s) for all fixations within an area of interest for all test stimuli throughout the experiment.

Fixation count measures the number of fixations in each area of interest for all test stimuli throughout the experiment. If during the recording the participant leaves and returns to the same media element, this is counted as a new fixation. A detailed description of the metric measures can be found in the Tobii Studio user manual (https://www.tobiipro.com/siteassets/tobii-pro/user-manuals/tobii-pro-studio-user-manual.pdf, accessed on 2 January 2019).

We exported the gaze data from Tobii Studio to SPSS [34] for further analysis. Statistical tests on the gaze data were preceded by a normality check on the distributions of the respective residuals by means of a Shapiro–Wilk test. In case normality could not be assumed, non-parametric tests were performed (Mann–Whitney and Wilcoxon tests).

In order to investigate the association between the behavioral data and eye movements, we performed Spearman correlations. We computed correlations between d’ (IR and DR) and eye tracker data (total fixation duration and fixation count) during encoding for both groups separately (young adults and older adults).

## 3. Results

### 3.1. Behavioral Results

LMM analysis on d’ revealed a significant interaction between group and emotion for both the IR and DR phase (all ps < 0.05, Bonferroni-corrected; Figure 4a). The interaction during IR revealed a higher d’ for angry faces compared to neutral faces (*p* < 0.001) for the young adults group but not for the older adults group (*p* = 0.514). Furthermore, compared to the older adults group, the young adults group showed a higher d’ for angry faces (*p* < 0.001, 48% difference) but not for neutral faces (*p* = 0.103). LMM further revealed a main effect of group (all ps < 0.001), with a higher d’ for the young adults group and a main effect of emotion, with a higher d’ for angry faces compared to neutral faces, for both IR and DR (all ps < 0.013).

In line with the results during the IR phase, the group x emotion interaction during DR revealed a higher d’ for angry faces compared to neutral faces (*p* < 0.001) for the young adults group but not for the older adults group (*p* = 0.796). Furthermore, the young adults group revealed a higher d’ for both angry (51% difference) and neutral (32% difference) faces compared to the older adults group (*p* < 0.001).

LMM for IR and DR with group, category, and group x category as fixed effects revealed no significant interaction (all ps > 0.362). The results revealed a main effect of group for IR (*p* < 0.001) and DR (*p* < 0.001) but not for category (all ps > 0.310; see Figure 4b).

Secondly, LMM analyses investigating the effect of interval revealed a significant interaction between group and interval for both the house and neutral face condition (all ps < 0.035, Bonferroni-corrected). The interaction was qualified as a higher d’ for DR compared to IR (all ps < 0.001) for the young adults group. This effect was not significant for the older adults group (all ps > 0.162). No interactions were observed between group and interval for the face and angry face condition (all ps > 0.143). All LMM analyses revealed a main effect of group (all ps <.001) and interval (all ps <.005) with a higher d’ for DR compared to IR. In addition, we performed Pearson correlations between performance in immediate recognition and performance in delayed recognition (Table 2).

### 3.2. Eye-Tracking Data

Heatmaps revealed task-compliant fixation on the fixation cross for both groups (young adults and older adults; Figure 3c,d). In addition, we created group-level heatmaps during the presentation of faces and houses using default eye tracker settings (Figure 3c,d). Each heatmap was created by adding the values whenever a fixation shares the same X and Y pixel location. Each point was color-coded. The radius was set at 50 pixels, corresponding to a total kernel of 100 pixels. A detailed description of the metric measures can be found in the Tobii Studio user manual (https://www.tobiipro.com/siteassets/tobii-pro/user-manuals/tobii-pro-studio-user-manual.pdf, accessed on 2 January 2019).

The Shapiro–Wilk test revealed that not all residuals were normally distributed). For the purpose of uniformity of analyses, we performed nonparametric tests on all eye-tracking data. Mann–Whitney U tests revealed a significantly higher total fixation count for the older adults group regardless of category (all ps <0.001; Figure 5a) or emotion (all ps < 0.001). For the areas of interest, only the eyes showed a similar significance (eyes, *p* = 0.008; angry eyes, *p* = 0.004; and neutral eyes, *p* = 0.020). In contrast, total fixation duration was lower in the older adults group compared to the young adults group and revealed a significant group difference for mouth (*p* = 0.006), angry mouth (*p* = 0.012), noses (*p* = 0.002), angry noses (*p* < 0.001), and neutral noses (*p* = 0.002). An overview of all data is given in Table A1 in Appendix A.

In addition, we conducted the Wilcoxon signed-ranks test to study the effect of category and emotion within both groups. Total fixation duration for emotion revealed a significant effect in the older adults group (*p* = 0.004), with longer total fixation duration for neutral faces ($\bar{x}$ = 53.29) compared to angry faces ($\bar{x}$ = 52.47); see Figure 5b. To follow-up on these results, we calculated the difference in total fixation duration between angry and neutral faces and performed a Wilcoxon signed-ranks test to study the group difference. The results revealed a significant larger difference in total fixation duration (angry faces–neutral faces) for the older adult group ($\bar{x}$ = 0.8155) compared to the young adult group ($\bar{x}$ = 0.1585). Total fixation duration for category was only significant in the young adults group (*p* = 0.019), resulting in longer total fixation duration for faces ($\bar{x}$ = 112.15) compared to houses ($\bar{x}$ = 110.29).

In the young adults group, Spearman correlation analyses revealed a positive correlation between total fixation duration and d’ for faces at IR (r = 0.489, *p* = 0.029) and DR (r = 0.671, *p* = 0.001) (Figure 6a,b). The older adults group only showed a positive significant correlation for d’ for faces during DR (r = 0.495, *p* = 0.027). Interestingly, both groups showed a positive correlation for angry faces but only with d’ at DR (young adults: r = 0.618, *p* = 0.004; older adults: r = 0.496, *p* = 0.026); see Figure 6c. For the young adults group, this was also the case for neutral faces (r = 0.570, *p* = 0.009). We did not find any correlations between fixation count during encoding and d’ (*p* ≥ 0.202).

Regarding the areas of interest, we observed a positive correlation between total fixation duration for nose during encoding and d’ during DR but only in the young adults group (r = 0.526, *p* = 0.017; Figure 6d). The results also revealed a positive correlation between fixation count for nose during encoding and d’ during IR for the young adults group (r = 0.574, *p* = 0.008).

## 4. Discussion

In the present study, we addressed age effects in the emotional enhancement of memory following consolidation for facial and emotional memory and associated eye-scanning patterns. The findings support the idea of a negativity enhancement effect in young adults.

### 4.1. Emotional Enhanced Memory (EEM)

The results reveal enhanced memory for negative facial expressions in the young adults group, but not in the older adults group, for both IR and DR. In addition, angry faces showed the largest group difference in memory, with nearly a 50% difference in d’ for both the IR and DR. However, these findings need to be interpreted with caution as no positive face stimuli were shown. Therefore, the observed negativity enhancement effect could be part of an overall enhancement effect of emotional stimuli. In neither of the groups did we observe evidence for facial enhanced memory.

These findings are in line with the previous literature revealing a negativity enhancement effect for young adults but not in older adults [35,36,37]. A previous study observed enhanced memory for faces with negative expressions compared to both positive and neutral faces in young adults, while this effect was absent in older adults [38]. Previous research related this age difference in emotional memory with changes in brain function and personality traits by aging. Lower amygdala activity in older subjects has been associated with poorer memory for negative faces [39,40,41]. On the other hand, a previous study observed that for older adults, personality traits influenced memory for faces but not for younger adults. A lower negative mood was associated with better recognition of positive and negative faces [38]. In addition, lower recognition accuracy for negative faces could be predicted by higher scores of extraversion and openness in older adults [38]. These findings indicate that for older, more open, and extraverted adults, negative stimuli may be less noticeable or may even be avoided [42].

This idea complies with the observed eye-scanning patterns, which reveal different total fixation duration for emotional faces between both groups. Whereas the older adults group showed a lower total fixation duration for angry faces compared to neutral faces, this effect was not significant in the young adults group. These results indicate that negative faces draw less attention or even evoke avoidance in the older adults group compared to the neutral faces, while there was no significant difference in fixation duration between angry and neutral faces in the young adults group. These findings are also in line with previous research stating that older adults look less at negative stimuli and more at positive stimuli [43,44]. In addition, we observed a positive correlation between total fixation duration and d’ for angry faces for both groups but only during DR.

In sum, the present findings reveal increased performance for negative stimuli compared to neutral stimuli only in the young adults group. In addition, young adults memorize negative stimuli, but not neutral stimuli, better than older adults.

### 4.2. Overall Memory Decline and Eye-Scanning Pattern Association

As we expected, behavioral results showed a group effect for d’, revealing an overall higher d’ for the young adults group compared to the older adults group. This is consistent with known age-related effects in face memory [20,21,22]. Interestingly, for the face stimuli, IR revealed a group difference for angry faces only. On the other hand, DR revealed a group difference for angry and neutral faces. These findings are in line with a previous study, observing more distinct patterns in memory performance between a middle-aged group and an older adults group for delayed verbal memory than for immediate verbal memory [45]. Table 2 reveals a positive correlation between performance in immediate recognition and performance in delayed recognition for both groups, indicating that to some extent the effects observed in delayed recognition could be attributed to the participant’s immediate recognition performance. A possible explanation for the overall memory difference could be related to the observed difference in fixation count. For the older adults group, fixation count was significantly higher regardless of category or emotion, while there was no observed difference in total fixation duration between both groups. This observation indicates that the older adults group maintains more fixations and eye movement transitions when encoding a face, resulting in a decreased average duration of each fixation. These findings are in line with the study of Firestone et al. (2007), who also reported more eye movement transitions between face features for older adults [10].

Previous work has shown the importance of creating associations between individual features for visual memory. Binding provides the concept that certain features belong together [46]. Especially for face memory, there is evidence for holistic face processing, binding individual features into a single representation [47]. Previous research has revealed that older adults have more difficulty than young adults in feature binding during memory tasks [46,48], resulting in a deficit in associating individual features into a lasting representation [49]. In addition, this age-related binding decline is thought to not only influence the memory for faces but also the processing of these features during encoding, i.e., eye-scanning patterns [10]. A previous study suggested that eye movements play a functional role in determining relations among face features during encoding [9]. This indicates that a binding deficit could lead to an increase in number of fixations to compensate for the reduced face processing, reflecting a change in the binding process for the older age group.

We observed a significant correlation between total fixation duration and d’ for faces in the young adults group during IR and for both age groups during DR. These findings suggest that total fixation duration is a strong predictor of d’ and indicate that the association between total fixation duration and d’ strengthens over time. This effect was also observed for total fixation duration and d’ for angry faces.

An alternative hypothesis for the group difference in fixation count could be due to the preference to attend to context in older adults. Context is known to influence eye-scanning patterns. Previous research showed that older adults attended more to context and that an absent context leads to poorer recognition for emotional faces in older adults [50]. The preference and possible search for context could explain the many fixations outside the stimuli for the older adults group. Figure 7 represents a cluster visualization, a graphic representation of areas with high concentrations of gaze data points (clusters). The current clusters display the data of all participants (100%) that have contributed with gaze data to the cluster.

### 4.3. Eye-Scanning Patterns within the Areas of Interest

Before data acquisition, we created three areas of interest: mouth, nose, and eyes (Figure 3a). These areas of interest were based on previous research showing fixation on these features [51,52]. Furthermore, eye-scanning patterns of young adults indicate that the encoding of individual features may be important for face memory performance [53].

We observed group differences in eye-scanning patterns within the areas of interest and observed significantly more total fixation duration for nose (total nose, angry nose, neutral nose) and mouth (total mouth and angry mouth) in the young adults group. Furthermore, total fixation duration for nose was positively correlated with d’ for face during DR but only for the young adults group.

The findings are in line with previous research showing that for young adults, fixating on the nose made the difference between remembered versus forgotten faces [10]. Different studies describe the area between the eyes and the nose as the most informative area of the face leading to the best memory performance [54,55]. In addition, in a study by Hsiao and Cottrell (2008) the nose was the most frequently viewed area for face recognition [56]. These findings are in line with previous research, emphasizing the importance of where individuals look on faces and the corresponding differences in how well people can recognize faces [54,57].

### 4.4. Limitations and Future Directions

A possible limitation of the current study is that we used a fixed fixation cross. Research has shown that the start position of the fixation cross influences eye-scanning patterns during face processing [58]. One could argue that the observed group difference in total fixation duration for nose is due to better initial fixation on the fixation cross (which was positioned around the nose) for the young adults group. However, the heatmaps (Figure 3c,d) reveal task-compliant fixation on the fixation cross for both groups. The fixation cross may have had a negative effect on eye-scanning patterns, not allowing participants to use the optimal individual fixation strategy in order to memorize the stimuli.

A second limitation of the current study is that we did not let the participants rate the angry faces in order to study arousal; this would be an interesting addition as older adults rate both negative and positive emotional faces as more intense compared to young adults [59].

A third limitation of the current study is the small sample size, potentially leading to a lower power. Our sample size, however, is comparable to previous research conducting eye-tracking or long-term memory studies with face stimuli, with sample sizes ranging from 15 to 37 participants per age group [60,61,62,63].

In the current study, we compared eye-scanning patterns for neutral versus angry faces. It would be interesting to add positive face stimuli (e.g., happy-faced) and compare different eye-scanning patterns for positive, negative, and neutral faces between different age groups in order to study the positivity effect, especially as there have been inconsistent findings regarding enhanced memory for positive stimuli [8,64]. In addition, the observed negativity enhancement effect could be part of an overall enhancement effect of emotional stimuli.

## 5. Conclusions

The present study supports a negativity enhancement effect for memory in young adults but not in older adults. The findings reveal enhanced memory for angry faces in the young adults group but not in the older adults group. The absence of negative emotion enhanced memory in the older adults could be explained by less attention to or avoiding negative faces, indicated by shorter fixations on angry faces compared to neutral faces.

## Figures and Tables

**Figure 1 brainsci-12-01719-f001:**
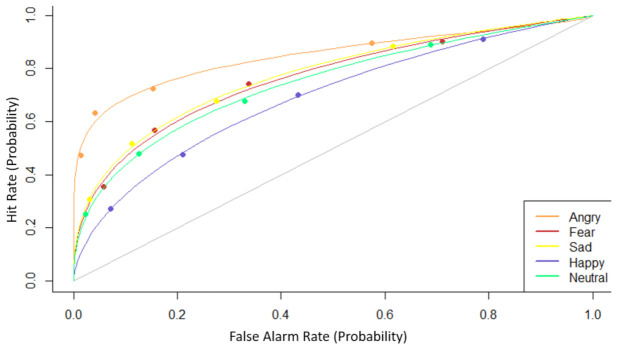
ROC curves for recognition of emotional facial expressions. Participants recognize angry faces best and the recognition for angry faces deviated the most from the recognition of neutral face stimuli.

**Figure 2 brainsci-12-01719-f002:**
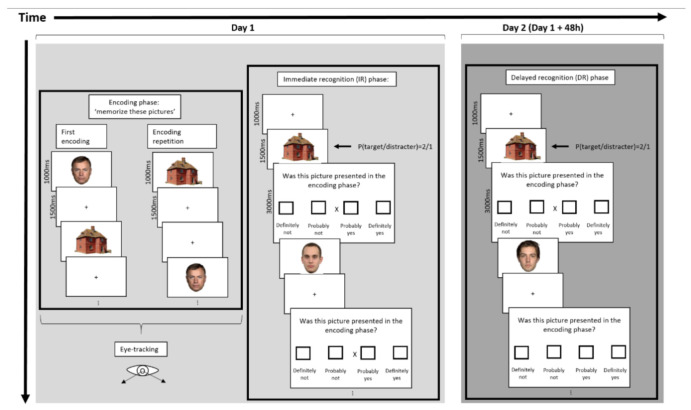
Schematic design presentation of the protocol. The experiment consisted of an encoding phase, an immediate recognition phase (IR), and a delayed recognition phase (DR). Eye movement data was collected during the encoding phase. The encoding phase (**left panel**) consisted of two sessions. In each session, 80 stimuli (40 houses and 40 faces) were pseudo-randomly presented. The second encoding session (encoding repetition) was identical to the first except for the order of stimulus presentation. Prior to the encoding phase, participants were instructed to memorize the stimuli of the encoding phase in order to recognize them in a subsequent recognition test. The IR phase (**middle panel**) directly followed the encoding phase. In the IR phase, the stimuli consisted of the 80 stimuli from the encoding phase intermixed with an additional 40 distractors: 20 houses and 20 faces. The trials were equally divided over 4 sessions during the IR phase. The response screen displayed the question “Was this picture presented in the encoding phase?” with four boxes below referring to four response alternatives: “definitely not”; “probably not”; “probably yes”; “definitely yes”. The DR (**right panel**) is conducted two days after the first session (48 h). The DR is identical to the IR, except for the stimulus presentation order and the distracter stimuli, i.e., 40 new distractors were presented in the DR.

**Figure 3 brainsci-12-01719-f003:**
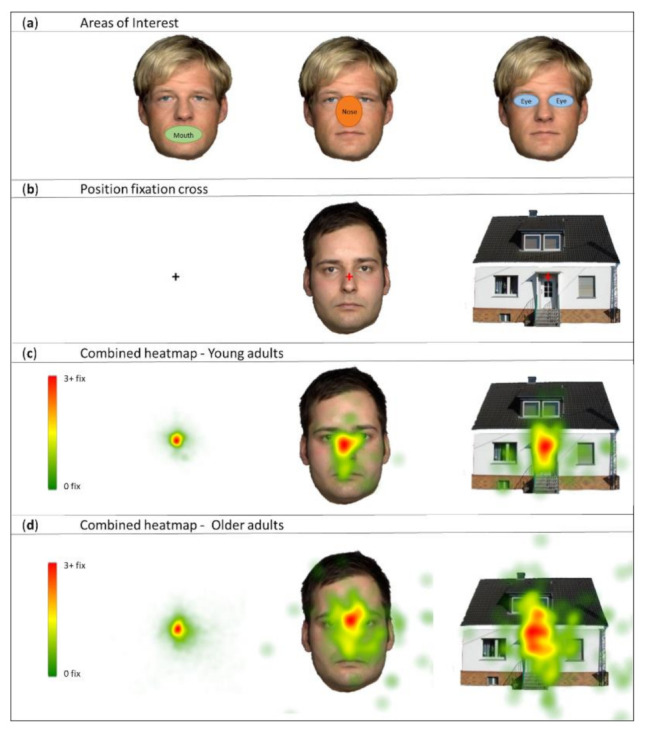
Position and heatmap fixation cross. (**a**) Areas of interest were created within Tobii Studio: mouth, nose, and eyes. (**b**) Red fixation cross illustrates where the fixation cross was positioned relative to the stimuli. (**c**) Group level heatmaps of test stimuli for the young adults group during presentation of 3 conditions: fixation, faces, and houses. (**d**) Group level heatmaps of test stimuli for the older adults group during presentation of 3 conditions: fixation, faces, and houses.

**Figure 4 brainsci-12-01719-f004:**
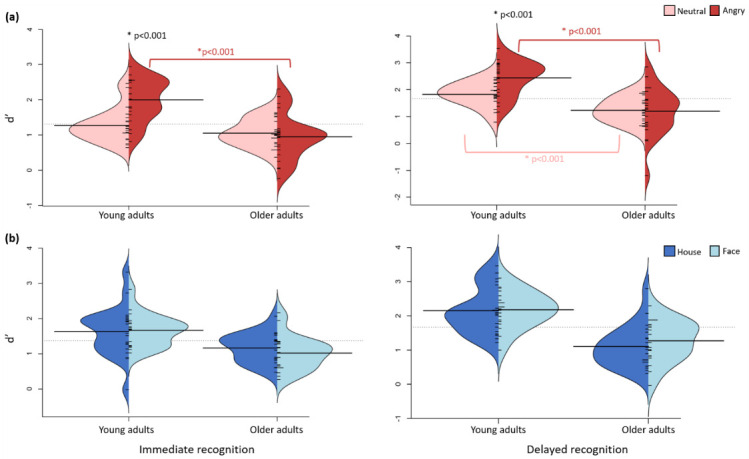
Split violin plots presenting the d’ for emotion and category. (**a**) Split violin plots displaying d’ as a function of emotion, group, and interval. In the young adults group, d’ was significantly higher for angry faces compared to neutral faces for both IR and DR (all ps < 0.001). (**b**) Split violin plots displaying d’ as a function of category, group, and interval. The results reveal a main effect of group for both IR and DR (all ps < 0.002). * *p* < 0.05.

**Figure 5 brainsci-12-01719-f005:**
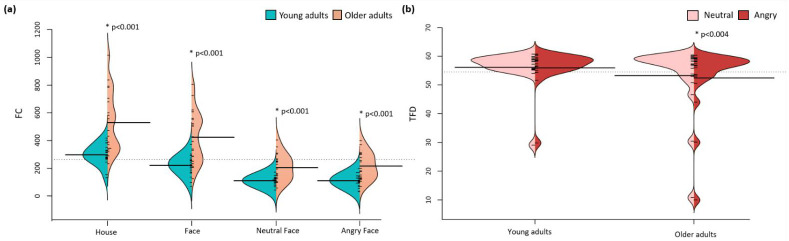
Split violin plots representing the eye-tracking data. (**a**) Fixation count for house, face, neutral face, and angry face conditions. Fixation count was significantly higher in the older adults group for all conditions (all ps < 0.001). (**b**) Total fixation duration for emotion. The results show a significantly higher total fixation duration for neutral versus angry faces (*p* = 0.004) within the older adults group. * *p* < 0.05.

**Figure 6 brainsci-12-01719-f006:**
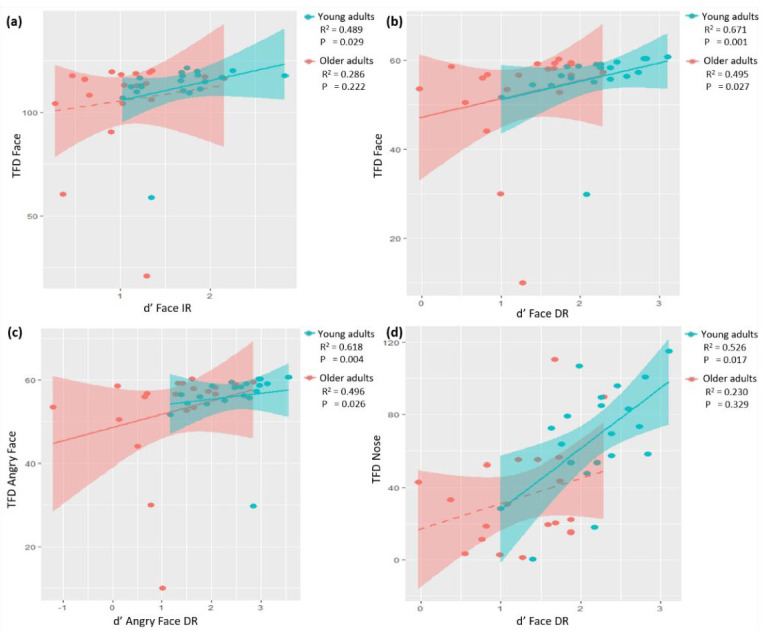
Scatterplots presenting the relation between total fixation duration (TFD) and d’s. (**a**) Correlation between total fixation duration (TFD) for faces and d’ for faces in IR, where only the young adults group reached statistical significance (r = 0.489, *p* = 0.029). (**b**) Correlations between total fixation duration (TFD) for faces and d’ for faces in DR are significant in both groups (young adults: r = 0.671, *p* = 0.001; older adults: r = 0.495, *p* = 0.027). (**c**) Correlations between total fixation duration (TFD) for angry faces and d’ for angry faces in DR are significant in both groups (young adults: r = 0.618, *p* = 0.004; older adults: r = 0.496, *p* = 0.026). (**d**) Correlations between total fixation duration (TFD) for nose and d’ for faces in DR, where only the young adults group reached statistical significance (r = 0.526, *p* = 0.017). Solid lines represent a significant correlation; dashed lines represent a non-significant correlation.

**Figure 7 brainsci-12-01719-f007:**
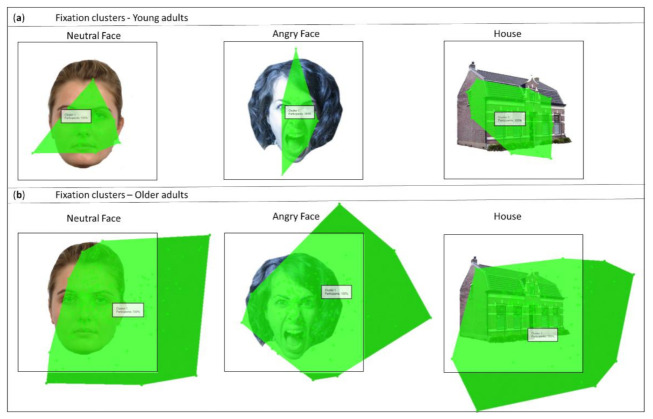
Graphic representation of areas with high concentrations of gaze data points (clusters). (**a**) Fixation clusters for the young adults group. From left to right: neutral face, angry face, and house. (**b**) Fixation clusters for the older adults group. From left to right: neutral face, angry face, and house.

**Table 1 brainsci-12-01719-t001:** Demographic data.

	Young Adults (N = 20)	Older Adults (N = 21)	*p*
Sex (♂/♀)	7/13	9/12	0.606
Age	21.6 (2.37)	69.3 (7.61)	<0.001 *
Age education	21.4 (2.46)	17.0 (3.06)	<0.001 *
ACE-III ^1^ (/100)	92.8 (4.67)	88.5 (6.66)	0.024 *
MMSE (/30) *	29.7 (0.67)	29.1 (1.07)	0.060

^1^ ACE-III = Addenbrooke’s Cognitive Examination III; MMSE = Mini–Mental State Examination. * significant group differences (*p* < 0.05).

**Table 2 brainsci-12-01719-t002:** Correlations between immediate recognition and delayed recognition.

		Young Adults (N = 20)	Older Adults (N = 21)
d’ Angry face	R^2^ *p*	0.669 0.001 *	0.481 0.032 *
d’ Neutral face	R^2^ *p*	0.295 0.206	0.024 0.920
d’ Face	R^2^ *p*	0.502 0.024 *	0.213 0.366
d’ House face	R^2^ *p*	0.736 <0.001 *	0.694 <0.001 *

* = significant group differences (*p* < 0.05).

## Data Availability

The data presented in this study are openly available in Dryad digital repository at https://doi.org/10.5061/dryad.3j9kd51p2, accessed on 5 December 2022.

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
