# Peer review of "Age Effects in Emotional Memory and Associated Eye Movements"

_brainsci, 2022, doi:10.3390/brainsci12121719_

Round 1
Reviewer 1 Report (New Reviewer)
I don't have any further suggestions
Reviewer 2 Report (Previous Reviewer 2)
I have no remaining comments.
Reviewer 3 Report (Previous Reviewer 1)
This is a revision of a paper I had reviewed during the initial submission.
The letter clearly describes the changes the authors have made in response to my comments, and I believe they have satisfactorily addressed most of my concerns. I was not able to find any further serious concerns.
This manuscript is a resubmission of an earlier submission. The following is a list of the peer review reports and author responses from that submission.
Round 1
Reviewer 1 Report
Summary:
This study aimed at specifying the mechanisms underlying emotional face recognition and social memory for young and older adults using both behavioural and eye tracker paradigms. As a whole, it was found that memory performance was enhanced for angry faces in the young adults but not in the older adults. Moreover, fixation durations for angry faces were shorter for the older adults as compared to neutral faces. The authors argued that the findings are consistent with the socioemotional selectivity framework.
General comment:
Overall, the study addresses an important issue. Clarifying the processes underlying emotional face recognition is interesting and valuable. Furthermore, the use of behavioural data in relation with eye tracker data is a strength of this study. However, the paper suffers from some shortcomings that the authors should address before the paper may be considered for publication. My main points concern the notion of social memory and the way it is considered in the paper, as well as the methodology used. As explained below, I think that the concept of “social memory” should be better defined and considered throughout the paper since it is mentioned as one objective of the present study. Also, more information should be provided concerning several important elements of the method (characteristics of the participants, the materials, the procedure) which could perhaps allow a better understanding of the results obtained.
Major points:
- In the title, it is mentioned that the paper will deal with emotional memory. I would rather say that it is here about emotional face recognition. Then, in the paper, it is indicated that one of the objectives is to study social memory. Finally, in the discussion, findings about social memory performance are not really discussed. So, I think that all these elements are a little bit confusing. If one objective is to study ‘social memory’ (which I think is clearly different from emotional memory), it should appear in the title. Also, it is important to have a clear definition of what “social memory” is, a presentation of the previous findings in the literature about this memory system, and a discussion of the present findings in relation with the literature. In the discussion it is simply stated that “in neither of the groups, we observed evidence for social enhanced memory”. More discussion is needed here.
- We do not have sufficient information in the method section. First, important information about the participants are missing, such as the level of education, the vocabulary level, performance in speed processing, ... These are critical information necessary to report since young and older adults generally (but not always depending on the selected sample) differ on these characteristics and this could impact the results. In fact, some previous studies have reported that the positivity effect was higher for older adults with sub-clinical memory deficits or with diminished cognitive resources comparable to those of individuals with Alzheimer’s disease (e.g., Bohn et al., 2016; Leal et al., 2016). A recent study has also indicated that a preference for negativity of small or even zero magnitude would be an indicator of good maintenance of cognitive resources during ageing (Laulan et al., 2021). Could the responses to the items at the ACE-III be used to provide more detailed information about the participants? (if both age groups completed the test). Moreover, it is important to have information about the emotional state of the participants (such as alexithymia, anxiety, affect scores) since some previous studies have shown that the memorization of emotional items depends on individuals emotional state, such as alexithymia (e.g., Vermeulen & Luminet, 2009). By the way, could you please specify whether vision was corrected or not for the participants (for the young and older adults). Finally, the age range was very high for the sample of older adults: Is it appropriate to consider the performance of 50-year-old participants and 90-year-old participants within the same group of old people? Middle-aged participants are mixed with old and very old participants that drastically differ in their memory abilities, and this is especially critical here since the size of the sample is quite small (n=20). Do you have the same pattern of data when you only consider adults aged 70 to 90?
- Also, in the material sections, more information should be provided. First, can you clarify the way the 200 images from the pilot study was selected? What are their mean valence scores and arousal scores? The same applies for the final 80 pictures. Since the study focuses on emotional factors, a critical information to give is the mean valence and arousal for both item conditions (houses, neutral and angry faces). Furthermore, did the pictures used correspond to faces of young and/or of older adults? Did you ensure that there were no differences on the evaluation of the material between younger and older adults (It seems that the pilot study was run of 13 young adults only)?
- In the procedure section, it is indicated that two encoding sections were used. Can you clarify and justify the use of this procedure since it seems quite unusual to have two encoding sections? Also, there were 40 foils in the recognition task with 80 old items. Can you explain the reason why there were not the same number of old and new items? As well as the potential impact of such proportion of foils?
- In the results, did you look at correlations between performance in immediate recognition and performance in delayed recognition? I think it would be interesting to have such information since it is unclear to what extent the effects observed in delayed recognition could be attributed to participant’s immediate recognition performance. The very act of retrieving and successfully recognizing the items could serve as an additional encoding for these items. In other words, those items that were recognized in immediate recognition would have enhanced encoding as compared to those items that were not recognized. Do you think that this could explain the fact that the correlations you found between total fixation durations and d’ for faces during recognition were higher for delayed than for immediate recognition?
Minor points:
- Please, avoid the use of abbreviations which are too numerous here and hinder understanding, particularly in the results section.
- Please correct the typos (e.g., p. 2, “positivity’ instead of “possitivity’)
Reviewer 2 Report
Many previous studies have shown differences in the effects of valence on attention and memory for younger vs. older adults. In this study, the authors aim to examine the links between these attention and memory effects.
Although this general question is interesting and worth pursuing, I think there are some significant problems with the design and execution of the present study. Perhaps most importantly, the lack of a positive valence condition means the study cannot test the hypotheses or support the conclusions that the authors spend much of the paper discussing. I also think there are a number of issues in the statistical analysis that need to be clarified or corrected. I elaborate on these and other concerns below.
MAJOR ISSUES
1. The authors need to be clearer about how they are defining the negativity bias and positivity effect. In particular
a) “However, less is known about the negativity bias in older adults.” I’m confused by this sentence. As the authors have just reviewed leading up to this, lots of research has been done looking at emotional biases in older adults. A relatively consistent finding is that older adults show either a bias for positive stimuli or a reduced negativity bias compared to younger adults. In fact, the positivity effect has been explicitly described as the fading of the negativity bias with age (Carstensen & DeLiema, 2018). If the authors think that the negativity bias and positivity effect are distinct and independent phenomena (as they also imply in line 87-88), they need make clear how they are defining these effects.
b) “A previous study, where older adults had to remember a list of words with negative, positive, and neutral valence, reported no evidence for a positivity effect, however, they did observe a reduced negativity effect in memory in older adults.” Again, the terminology is unclear here. In much of the literature, a reduced negativity bias in older adults would be considered an example of the positivity effect. The positivity *effect* does not require older adults show a positivity *bias*. A number of factors (other than valence alone) will determine which stimuli receive more attention and are better remembered, so it is not surprising that that are some paradigms or stimulus sets where older adults remember more negative than positive. But if this difference is larger in younger adults (a larger negativity bias), that still fits the pattern of the positivity effect.
2. “Therefore, in the current study, we only focus on negative versus neutral stimuli to observe negative bias in more detail.”. This does not actually allow for examination of the negativity bias. If negative stimuli are better attended or remembered than neutral stimuli, that can be explain by a general enhancement of attention and memory by emotion. Examining a specific bias for negative requires the inclusion of positive stimuli that are otherwise well-matched to negative (e.g., in arousal). In the present study, the authors find that younger adults show NEG>NEU in memory but older adults do not. But without a positive condition, it is impossible to know whether this reflects a reduced negativity bias in older adults or a reduced effect of emotion/arousal. Thus I’m not sure the design can support the tests of hypothesis and conclusions the authors want to relate it to.
3. The hypotheses need to be clearer:
a) “Furthermore, we expected enhanced memory for faces with negative expressions (negativity bias) in the young adults group, but not in the older adult group.” I do not believe SST makes this prediction. SST’s prediction of a positivity effect is consistent with a general emotional enhancement of memory such that negative will be remembered better than neutral by all age groups.
b) “This led to the hypothesis that eye-scanning patterns predict memory outcomes.” What led to this? It’s not clear how this follows from what comes just before.
c) “To further document the nature of SST, we analysed correlations between memory performance and eye tracker data, for both groups separately (young adults and older adults).” Does SST predict this? SST makes predictions for valence effects in attention and memory. I’m not sure it makes any particular prediction about how eye tracking should be correlated to memory. The authors should be clearer about how they derive their predictions from SST.
4. I think the authors need more justification/motivation for their inclusion of covariates.
a) “Independent Sample t-test showed a significant group difference for ACE-III (p=.024), for this reason, we include ACE-III as covariate of no interest in the analyses.” This needs more justification. Just because a variable differs between groups does not mean it should be controlled for. Adding ACE-III to the model means that all effects involving group examine whether there are any group differences when general cognitive ability is held constant. This may be an interesting question to ask, but it is a different question than the more general question of whether valence effects change with age, because cognitive decline is a normal part of aging. That is, the results with this covariate lack an aspect of external validity: They do not reflect the actual differences between young and old adults, they reflect the differences we would expect in a hypothetical world where cognitive function does not decline with age. Again, that is an an interesting question in it’s own right, but the authors need to motivate this choice. Adding ACE-III as a covariate of no interest also implies that cognitive functioning is a confound rather than, for example, a mediator. This perspective also requires justification. Finally, I would note that ACE-III was included as covariate in the LMMs, but not the non-parametric analysis. That makes the interpretation importantly different between these different analysis, but this is not justified or addressed.
b) Similarly, in the Results section we learn that sex was included as a covariate in at least some models. This is never justified and there were only small gender differences between the groups. In general, adding more predictors to a regression model decreases the power and estimation precision for all terms in the model, and also adds additional elements to the interpretation, so all terms in the model should be justified.
5. The key hypotheses are about interaction effects, but many of the key analysis are not conducted in a way that can support conclusions about interactions.
a) “In case normality could not be assumed, non-parametric tests were performed.” Non-parametric tests are a large family of tests, so the authors need to be more specific. Based on the Results section, the authors seems to only use Mann-Whitney and Wilcoxon tests. However, the key questions posed in their introduction are about Group x Valence interactions. Tests comparing groups within a condition or conditions within a group cannot test these hypotheses (see Gelman & Stern, 2006; Nieuwenhuis et al., 2011). It’s also worth nothing that while these tests do not assume normality, the Mann-Whitney test does assume equivalent distributions between conditions, an assumption that is frequently violated (e.g., by unequal variances or skew between groups/conditions). See Wilcox (2021) for a detailed treatment of robust alternatives to normal model tests and for how to test interactions with tests that do not assume normality.
b) Similarly, the Spearman correlations within each group cannot test hypotheses about differences between the groups. This requires a model that directly tests an interaction effect (Gelman & Stern, 2006; Nieuwenhuis et al., 2011).
OTHER ISSUES
6. From the abstract: “Twenty 14 young adults and 24 older adults encoded stimuli depicting neutral faces, angry faces, and houses 15 twice, while eye movement data was collected.” What is meant by “twice” here? The Methods describes seems to describe a procedure where each stimulus is only encoded once.
7. “A previous study revealed evidence for SST in long-term memory [31]” A number of studies have examined the predictions of SST in LTM. It’s not clear why this study is being singled out here.
8. “A power analysis was calculated using G*Power [33] with d = 1.23, an α error probability of 0.05, and a β value of 0.95, which estimated two groups of 19 and 19 participants respectively (N = 38). The predicted effect size for the current study is based on the study by Joubert et al. (2018)[4], who reported a negativity bias for young adults (d = 1.72) but not for older adults’ (d = 0.79).” There are a few issues here:
a) Most importantly, while I’m not familiar with the study of Joubert et al., d = 1.23 seems optimistic as an estimate of the effect size. This is a huge effect. In general it is risky to based estimate of effect size for power analysis on a single study with a small sample size, especially given publication bias against null effects. Reed et al.’s (2014) meta-analysis of 100 studies of the positivity effect revealed an overall average effect size of d = 0.257 (although this was for the comparison of negative and positive).
b) β is the Type II error rate. I assume the authors mean power (1 - β) = 0.95.
c) The last line is confusing, because how does d = 0.79 (a large effect) indicate no negativity bias for older adults?
9. For LMM models, the authors should make clear how the categorical variables were coded in the model (it is impossible to interpret main effects in the model without knowing this) and which random factors were included.
10. For the analyses with category as a factor, the Methods states that faces were compared to houses. If the authors averaged across all faces, how do they control for the valence confound in this comparison (i.e., half the faces are angry whereas presumably all the houses are neutral)? Is there a reason to do two different LMM models rather than just one Group x Condition (angry face, neutral face, house) model with pairwise follow-ups?
11. “We applied Bonferroni-correction for multiple comparisons.” What was considered a family of tests for these corrections? Was a multiple comparison correction also applied to account for the multiplicity of eye-tracking measures?
12. If I understand correctly, there were 10 eye tracking measures: TFD and FC for each of house, full face, mouth, nose, and eyes. But full results are not reported or graphed for all of these metrics for every contrast (for the most part, only the ones that reached significance are reported with stats). I think the authors should figure out some way of systematically reporting all contrasts for all metrics. A table that gives an effect size and p-value for each contrast would work well.
13. Line 407-408: “there are different underlying explanations for this theory”. This phrasing is confusing. Do the authors mean there are different explanations for the findings or for SST? I’m not sure what it means to say there are different explanations for SST; SST is an explanation for a set of empirical regularities.
14. The authors should read the paper closely for errors of grammar and spelling. As a couple of examples:
a) “Witch” should “Which” in line 61. This sentence is also not a complete sentence.
b) The sentence starting on line 85 is not a complete sentence.
c) “possitivity” should be “positivity” (line 88)
REFERENCES
Carstensen, L. L., & DeLiema, M. (2018). The positivity effect: A negativity bias in youth fades with age. Current Opinion in Behavioral Sciences, 19, 7-12. https://doi.org/10.1016/j.cobeha.2017.07.009
Gelman, A., & Stern, H. (2006). The difference between “significant” and “not significant” is not itself statistically significant. The American Statistician, 60(4), 328-331. https://doi.org/10.1198/000313006X152649
Nieuwenhuis, S., Forstmann, B. U., & Wagenmakers, E. J. (2011). Erroneous analyses of interactions in neuroscience: a problem of significance. Nature neuroscience, 14(9), 1105-1107. https://doi.org/10.1038/nn.2886
Wilcox, R. R. (2021). Introduction to robust estimation and hypothesis testing (5th Ed.). Academic Press. https://doi.org/10.1016/C2019-0-01225-3 .
Round 2
Reviewer 1 Report
This is a revision of a paper I had reviewed during the initial submission.
The letter clearly describes the changes the authors have made in response to my comments, and I believe they have satisfactorily addressed most of my concerns. I was not able to find any further serious concerns.
Note however that there is a substantial number of typos that have to be corrected by the authors.
Author Response
We thank Reviewer#1 for the appreciation of the manuscript and have corrected the revised manuscript for typos.

Reviewer 2 Report
1. I remain unconvinced that this study can address the questions the authors want it to. I refer back to my first review for details, but the simple issue is this: Most of the introduction discusses the positivity bias, then the authors present a study that does not include positive stimuli. It is simply not possible to test questions about the positivity bias, or questions about valence effects more generally, without including positive stimuli. The current study compares negative and neutral conditions that differ on general emotional relevance, arousal, and valence. Given that emotionality and arousal are known to have effects on both attention and memory—effects that are stronger and more consistently replicated than valence effects—any differences between these conditions is most sensible interpreted as an effect of emotion/arousal, and cannot support any claims about valence.
2. The authors still base their power analysis on a single study with a large effect size. They now note that other studies have used similar sample sizes, but this offers little support because we know low power is a common problem. As the author’s own review in the Introduction makes clear, the literature has found a wide range of effect sizes in other studies of emotion/valence effects on attention and memory, and not all studies have found differences between older and younger adults at all. Thus, a power analysis based on one study that did find large differences between age groups is not convincing. N=20 in each group in a between groups design only has high power with large effects sizes (most effects examined in psychology are not this big), and thus should be considered underpowered in the absence of a convincing argument that the expected effect size is large. Because low power also undermines confidence in significant effects, this is a serious concern, which is why there is a push to move to larger sample sizes in psychology, particularly in between subjects designs.
3. In my first review, I said that the inclusion of ACE-III and gender as covariates required justification. The authors respond to this in the response to reviewers but in the manuscript, they simple say “"as cognitive decline and gender are known to have an influence on memory performance". For reasons explained in my first review, this is not, in itself, a reason to control for these variables. The choice to control depends on the question being asked and how these third variables are conceptualized (e.g., as mediators, confounds, moderators). The authors need a theory/hypothesis driven motivation for their choice. There also remains the problem that the authors control for these variables in the memory analyses, but not the eye-tracking analyses. Given that a goal of the study is to related eye-tracking to memory, doing the analyses in way that requires interpreting results for these two different measures in conceptually different ways seems odd.
4. I’m afraid the authors misunderstood the point I made about testing interactions in eye-tracking. Take, for example, this part of the Discussion: "Whereas the older adults group showed a lower total fixation duration TFD for angry faces compared to neutral faces, the young adults group did not. These results indicate that negative faces draw less attention or even evoke avoidance in the older adults group, compared to the neutral faces, while this effect is absent in the young adults group." The fact that young adults did not show a significant effect does not prove that there is no difference in young adults (p>.05 doesn’t prove the null is true), and the fact that the effect is significant for one group and not the other does not prove there is a group difference—that requires a significant interaction (Gelman & Stern, 2006; Nieuwenhuis et al., 2011). The key questions of the study are about whether older and younger adults show different effects of emotion (i.e., negative – neutral). These questions require testing interaction effects, and the stats the authors present for the eye-tracking analyses do not do this.
5. In my first review, I noted that the coding for categorical variables in the LMM model needs to be clarified, as this has important implications for how effects in the models are interpreted. The values of coefficients depend on this, but, even more importantly, main effects in an interaction model have dramatically different interpretations depending on coding scheme. If, for example, dummy coding was used, then these main effects do not reflect what the authors are interpreting them to reflect. The authors respond by posting their SPSS syntax (although there is no link, so I was not able to view this syntax), but this is a key detail that readers should not need to find and interpret syntax to understand. The authors also need to make clear the random effects structure of the models.
6. I still don’t understand exactly what was Bonferroni-corrected. The authors state that they preformed correction “within the LMM”. Do they mean across the different predictors within a single model? Please be explicit.
REFERENCES
Gelman, A., & Stern, H. (2006). The difference between “significant” and “not significant” is not itself statistically significant. The American Statistician, 60(4), 328-331. https://doi.org/10.1198/000313006X152649
Nieuwenhuis, S., Forstmann, B. U., & Wagenmakers, E. J. (2011). Erroneous analyses of interactions in neuroscience: a problem of significance. Nature neuroscience, 14(9), 1105-1107. https://doi.org/10.1038/nn.2886
